# Combining photocatalytic hydrogen generation and capsule storage in graphene based sandwich structures

Li Yang[1,*], Xiyu Li[1,*], Guozhen Zhang[1,*], Peng Cui[1], Xijun Wang[1], Xiang Jiang[2,3], Jin Zhao[2,3], Yi Luo[1,3] & Jun Jiang[1]

The challenge of safe hydrogen storage has limited the practical application of solar-driven photocatalytic water splitting. It is hard to isolate hydrogen from oxygen products during water splitting to avoid unwanted reverse reaction or explosion. Here we propose a multi-layer structure where a carbon nitride is sandwiched between two graphene sheets modified by different functional groups. First-principles simulations demonstrate that such a system can harvest light and deliver photo-generated holes to the outer graphene-based sheets for water splitting and proton generation. Driven by electrostatic attraction, protons penetrate through graphene to react with electrons on the inner carbon nitride to generate hydrogen molecule. The produced hydrogen is completely isolated and stored with a high-density level within the sandwich, as no molecules could migrate through graphene. The ability of integrating photocatalytic hydrogen generation and safe capsule storage has made the sandwich system an exciting candidate for realistic solar and hydrogen energy utilization.

[1] Hefei National Laboratory for Physical Sciences at the Microscale, iChEM (Collaborative Innovation Center of Chemistry for Energy Materials), CAS Key Laboratory of Mechanical Behavior and Design of Materials, School of Chemistry and Materials Science, University of Science and Technology of China, Hefei, Anhui 230026, China. [2] ICQD/Hefei National Laboratory for Physical Sciences at the Microscale, and Key Laboratory of Strongly-Coupled Quantum Matter Physics, Chinese Academy of Sciences, and Department of Physics, University of Science and Technology of China, Hefei, Anhui 230026, China. [3] Synergetic Innovation Center of Quantum Information & Quantum Physics, University of Science and Technology of China, Hefei, Anhui 230026, China. * These authors contributed equally to this work. Correspondence and requests for materials should be addressed to J.J. (email: jiangj1@ustc.edu.cn).

Aiming to develop clean and sustainable energy resource, photocatalytic water splitting has gained great interests in both academia and industry communities[1–3]. By harvesting solar light, a photocatalyst generates energetic charges to split water into hydrogen ($H_2$) and oxygen ($O_2$) molecules[1], through which solar power is converted to hydrogen power, a clean and high calorific energy resource. Over years, many promising photocatalysts have been made, such as metal oxides or sulfides[4–7], metal organic frameworks[8], pure metals[9] and metal-free semiconductors[10] and so on. Recently, GR-based non-metallic photocatalysts exhibited advantages in both high performance and low cost[11–14]. For instance, g-$C_3N_4$, a graphitic carbon nitride, offers good photocatalytic performance together with thermodynamic stability and engineerability[11]. A recent work using nano-composite photocatalyst of g-$C_3N_4$ and carbon nanodot achieved solar energy conversion efficiency of 2.0% for water splitting[12].

Unfortunately, the future of widely utilizing hydrogen energy generated from sustainable solar and water resources is hampered by the difficulty of hydrogen collection and storage. Hydrogen generation relies on the delivery of photo-generated electron and hole charges to the photocatalyst reductive and oxidative sites, respectively. To enable energetic carriers driving reactions, the distance between reductive and oxidative sites is limited to the charge migration length (electron mean free path in 10–50 nm). Moreover, protons produced at oxidative sites need to migrate to reductive sites for $H_2$ generation, which also requires short reductive–oxidative distance. However, the short reductive–oxidative distance not only promotes unwanted reverse reactions (recombining proton and –$OH/O_2$ to water) that seriously reduce the efficiency, but also leads to extreme difficulties to collect or store $H_2$ without oxygen contamination. In practice, hydrogen storage is known to be a technical challenge. The mixture of $H_2$ with $O_2$ easily causes dangerous explosion, while metal or carbon fibre containers widely used for $H_2$ storage are expensive. Thus, the practical utilization of photocatalytic water splitting could not be realized until a cost-effective solution is developed to completely isolate hydrogen from oxygen during reactions and safely store $H_2$ afterwards.

Meanwhile, the technique advancements[15–17] in fabricating large-area GR and GR oxide (GO: GR modified by hydroxyl and epoxy groups) make the capsule of an appreciable amount of $H_2$ feasible. GR-based materials have exhibited excellent hydrogen-adsorption ability with relatively low cost compared with widely used metal materials. For instance, by combining palladium and activated carbon with GR, Lee and colleagues[18] achieved high performance in hydrogen storage capacity reaching 0.82 wt% (by weight) at ~8 MPa (improving nearly 49% compared with pure palladium). Urban and colleagues[19] constructed an environmentally stable GO and Mg nanocrystal composite with atomic thickness, which accomplishes hydrogen storage of 6.5 wt%.

Recently, hetero-structures based on two-dimensional (2D) nanosheets have also attracted great attention owing to excellent properties and fabrication feasibility[20,21]. Xie and colleagues[22] constructed a model of GR confined ultrathin tin quantum sheets, achieving high electrochemical activity and excellent stability for carbon dioxide radical anion. As a metal-free system, GR–$C_3N_4$ composite layers can effectively harvest visible solar light[14]. The GR-based materials hold high mobility in delivering charges towards efficient electrochemical and photochemical reactions[23,24]. Efficient electron–hole separation could be achieved through ultrafast charge transferring between GR–$C_3N_4$ layers despite of their weak van der Waals interaction, which has been experimentally observed in vertically stacked transition-metal dichalcogenide heterostructure bilayers with weak van der Waals

interaction[25–27]. Moreover, functional groups on GR, including hydroxyl and epoxy, doped heteroatoms, and defects, provide good active sites for water decomposition[20,28–32]. Our theoretical and experimental works demonstrate that parts of photo-generated electrons in carbon nitrides are collected at nitrogen positions[33,34], providing ideal reductive sites for hydrogen generation[35]. Moreover, Geim, Wu and colleagues[36] have demonstrated that GR film is not permeable for any particle larger than proton (or hydrogen atom), and the penetration of proton would be particularly efficient if driven by electrostatic attractions between protons and negative-charged ions.

Therefore, we can anticipate that a proper use of all these remarkable properties of GR-based materials can lead to a strategy to utilize the safe storage of hydrogen during the water splitting. For this purpose, here we design a multi-layer structure of $C_xN_y$ and GR network, where a graphitic carbon nitride (g-$C_xN_y$) layer is sandwiched between two GR layers modified by functional groups ($GR_F$). It takes advantage of the high efficiency of $GR_F$–$C_xN_y$ for photocatalytic water splitting, and the strict mass transport selectivity of GR to achieve safe capsule $H_2$ storage. Based on three known graphitic carbon nitride structures of CN, $C_2N$, $C_3N_4$ (Fig. 1a), we build the model sandwich system $GR_F$–$C_xN_y$–$GR_F$ (Fig. 1b, where g-CN and GO are used as prototype). The water splitting and hydrogen capsuling scheme is illustrated in Fig. 1b: (1) by absorbing visible or ultraviolet light, GO–$C_xN_y$ generates excitons, which soon separate to energetic electrons and holes. Specifically, electrons on the inner g-$C_xN_y$ mainly localized on N sites, while holes transfer to two outer GO layers, according to different material properties. (2) The holes on GO attack water molecules adsorbed on GO functional sites, triggering water splitting to produce protons ($H^+$). (3) Driven by electrostatic attractions, protons penetrate through GO to meet electrons on N sites of g-$C_xN_y$, and consequently produce $H_2$ molecules. (4) Since $H_2$ cannot transport through GR or $GR_F$ (GO), it would be retained in between two GO layers, realizing the purpose of capsule storage. In view of the above processes, a series of related calculations are carried out, including charge separations, chemical reactions, and $H_2$ storages. Our calculation results confirm the rationality and feasibility of integrating hydrogen production and capsule storage into one system. We study the electronic structures and couplings of GR network with g-$C_xN_y$ in the hybrid structures (Supplementary Fig. 1), to examine their stability, photo-absorption and charge distributions. Reactions of water splitting (oxidative) and hydrogen generation (reductive) are then investigated based on isolated monolayers of $GR_F$ sheet and g-$C_xN_y$, respectively. Hydrogen storage ability is studied with the GR–$C_xN_y$–GR sandwich model.

## Results

**Photo-generated electrons and holes separation.** For all optimized sandwich structures, g-$C_xN_y$ and GR/GO layers exhibit similar interfacial distances of 2.94–3.26 Å and adhesion energies from 1.02 to 3.73 eV (Supplementary Table 1), suggesting good structural stability. The interfacial distance and adhesion energy between g-$C_3N_4$ and GR are about 3.09 Å and 1.04 eV, agreeing well with the previous report[14]. The simulated electronic structures confirmed the semiconductor feature of g-$C_xN_y$, which enables photon energy harvesting. The computed dielectric function suggested that bare CN, $C_2N$, $C_3N_4$ mainly absorb ultraviolet light (Supplementary Fig. 2), in consistent with their band energy gaps (Supplementary Fig. 3). Their gaps were narrowed by coupling with GR/GO layers (Supplementary Fig. 3)[14], enabling the hybrid systems to harvest both visible and ultraviolet photons (Supplementary Fig. 2), and thereby

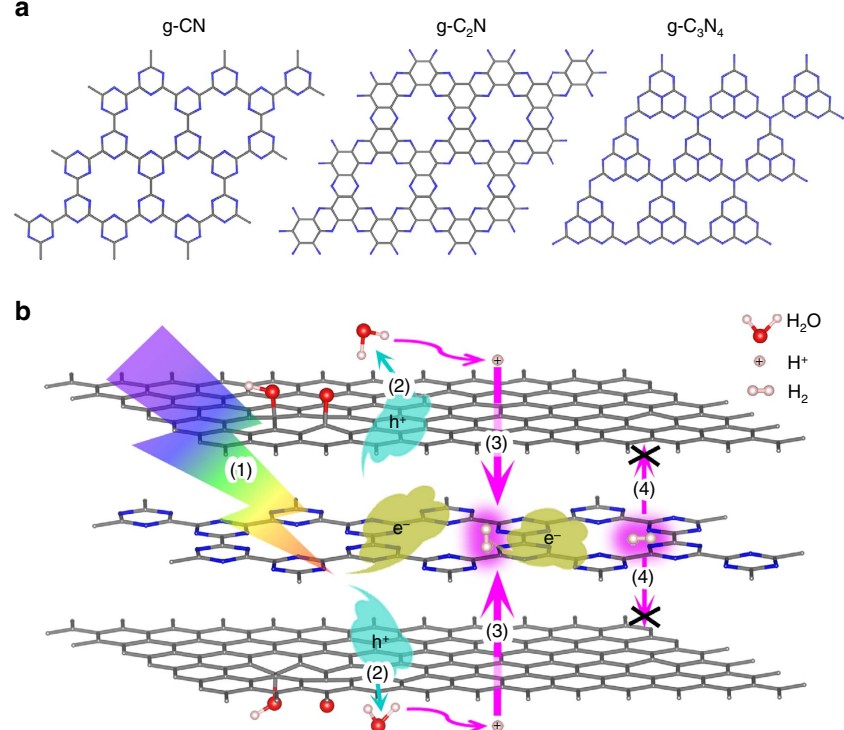

**Figure 1 | Scheme of water splitting and hydrogen capsuling scheme.** (**a**) Models of graphitic carbon nitrides CN, $C_2N$, $C_3N_4$. (**b**) The photocatalytic (water splitting) hydrogen generation and capsule storage scheme: (1) photo-generated electrons ($e^-$) and holes ($h^+$) separating; (2) water splitting to produce protons ($H^+$) through holes ($h^+$) attacking; (3) protons ($H^+$) penetrating through GO and producing $H_2$ molecules; (4) $H_2$ molecules are prohibited from moving out of the sandwich. Here GO–CN–GO is used as an example. Blue, grey, pink and red beads stand for N, C, H ($H^+$) and O atoms, the yellow and light blue clouds are for photo-generated electrons ($e^-$) and holes ($h^+$), and the blue and magenta arrows represent the migration of corresponding particles.

| Table 1 | Bader charge analysis of the sandwich structures. | | |
|---|---|---|---|
| **Neutral system GR/GO hole ($h^+$)** | **GR** | **$GO_{OH}$** | **$GO_O$** |
| GR/GO–CN–GR/GO | 0.34 | 0.72 | 0.54 |
| GR/GO–$C_2N$–GR/GO | 0.19 | 0.20 | 0.18 |
| GR/GO–$C_3N_4$–GR/GO | 0.22 | 0.25 | 0.16 |
| **$1e^-$ induced $C_xN_y$ electron ($e^-$)** | **CN** | **$C_2N$** | **$C_3N_4$** |
| GR–$C_xN_y$–GR | −0.68 | −0.45 | −0.33 |
| $GO_{OH}$–$C_xN_y$–$GO_{OH}$ | −0.79 | −0.33 | −0.32 |
| $GO_O$–$C_xN_y$–$GO_O$ | −0.87 | −0.41 | −0.32 |
| **$1h^+$ induced GR/GO hole ($h^+$)** | **GR** | **$GO_{OH}$** | **$GO_O$** |
| GR/GO–CN–GR/GO | 1.17 | 1.59 | 1.42 |
| GR/GO–$C_2N$–GR/GO | 1.01 | 1.10 | 1.01 |
| GR/GO–$C_3N_4$–GR/GO | 1.20 | 1.23 | 1.11 |

The computed charge distributions on GR/GO and $C_xN_y$ layers in the neutral sandwich systems, and systems with one extra (photo-generated) electron and hole carriers. Here $GO_{OH}$ and $GO_O$ represent the hydroxyl and epoxy GO, respectively.

convert the majority of solar power into energized electrons and holes.

The next key step is to separately deliver photo-generated electrons and holes to reductive and oxidative reaction sites. It is well documented that a difference in work function between different materials could induce electron flowing from the material with lower work function to the one with higher work function[37]. Here we found that the GR-based material has lower work functions than the $C_xN_y$, ranging from 1.7 to 2.91 eV (Supplementary Fig. 4). Therefore, before the photo-excitation, the sandwiched g-CN, $C_2N$ and $C_3N_4$ unit cells could donate 0.34–0.72, 0.18–0.20 and 0.16–0.25 positive charges ($h^+$) to the

outer GR/GO layers, respectively (Table 1, Supplementary Fig. 5), exhibiting well-separated electron and hole carrier distributions. We also carried out time dependent *ab initio* non-adiabatic molecular dynamics (AI-NAMD) simulations to describe the ultrafast hole evolution process. In Supplementary Fig. 6, it is shown that the hole with lower energy (near the valence band) could transfer from CN to GR, and ∼14% hole carriers would reach the GR sheet within 3 ps. While for the hole of higher energy, more than half of the carriers would quickly transfer to the GR sheet within 0.4 ps. Ultimately, above 80% hole carriers in CN layer would transfer to the GR layer within a few ps. Such an ultrafast charge transfer can compete with the electron-hole

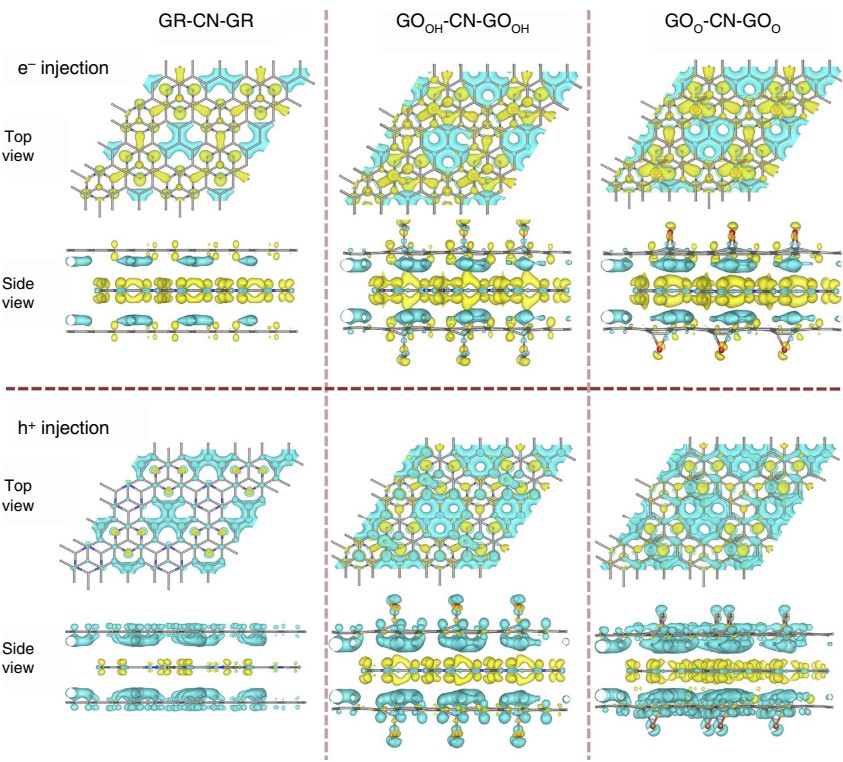

**Figure 2 | Photo-generated carrier distributions in the sandwich structures.** Charge distribution computed as Bader charge differences between GR/GO–CN–GR/GO sandwich with one extra carrier (photo-generated electron ($e^-$) or hole ($h^+$)) and the neutral monolayers of GR/GO and g-CN, from top and side view. Yellow and blue bubbles represent electron and hole charges with isosurface value of 0.0005 e Å$^{-3}$.

recombination and ensure the subsequent reactions (that is, water splitting on GO, and $H_2$ generation on $C_xN_y$). By adding 1.0 extra electron or hole carrier to the sandwich system, we have modelled the distributions of photo-generated charges. As displayed in Fig. 2 and Supplementary Fig. 7 and 8), the outer GR or GO layers always collect hole carriers, whereas inner g-$C_xN_y$ accumulates electrons. In Table 1, one photo-generated electron carrier induces about 0.68–0.87, 0.33–0.45 and 0.32–0.33 e$^-$ in the inner CN, $C_2N$ and $C_3N_4$ units, respectively. Although the GR/GO cells can collect 1.17–1.59 (with CN), 1.01–1.10 (with $C_2N$) and 1.01–1.23 (with $C_3N_4$) h$^+$ if one extra hole was injected, suggesting photo-generated holes on the outer GO layers.

**Water splitting driven by photo-induced energetic charges.** The water splitting starts with $H_2O$ adsorption to the outer $GR_F$ surfaces. Functional groups on GR induce polarized charges around active sites, which is helpful for $H_2O$ adsorption (Fig. 3a–c). Relatively high adsorption energies were found for water on various $GR_F$ materials ($E_{ads}$ in Supplementary Table 2), with 0.32–0.37 eV for GO (Supplementary Fig. 9), 0.60–1.02 eV for $GR_M$ (metal: Zn, Cu, Fe, Co, Ni in Supplementary Fig. 10), 0.47 and 0.07 eV for $GR_{Si}$ and $GR_N$ (Supplementary Fig. 11), 1.05 eV for $GR_{TiN4}$ (TiN$_4$ on GR in Supplementary Fig. 12), and 0.20 eV for $GR_{Cv}$ (GR with a carbon defect C$_v$ in Supplementary Fig. 13), respectively. Noticeably, it is shown in Fig. 3a–c that functional groups could collect much hole carriers after adsorbing water molecule, making it ready for oxidative reaction. Moreover, these groups effectively reduce the energy barrier ($E_b$) for water splitting. Based on climbing image nudged elastic band calculations for transition states (Supplementary Figs 9–13), the $E_b$ value of 5.13 eV for bare $H_2O$ is decreased to 3.64 eV with the

GR catalyst, 3.34–3.56 for the GO$_{OH}$ and GO$_O$ catalysts, 0.58–1.09 eV for the $GR_M$ (metal: Zn, Cu, Fe, Co), 0.41 eV for the $GR_{Si}$, 0.86 eV for the $GR_{TiN4}$ and 0.40 eV for the $GR_{Cv}$ (Supplementary Table 2). During photocatalysis, such energy barriers could be easily overcome by $GR_F$ receiving photo-induced energetic hole carriers. Similar to GO/GR, $GR_F$ with other functional groups tend to extract positive charges from the middle $C_xN_y$ layer (Supplementary Fig. 14 and Supplementary Table 3), validating the feasibility of photo-generated charge separation. The hole injection effectively reduced the Gibbs free energy for water splitting from 1.55 and 2.34 eV at the neutral state to 1.08 and 1.67 eV for the positively charged GO$_{OH}$ and GO$_O$, respectively, favouring water splitting in terms of thermodynamics (Supplementary Table 4).

**Protons penetration and hydrogen generation.** Driven by the electrostatic attraction force, protons produced at the oxidative sites would penetrate through the outer GR-based layers to meet the inner g-$C_xN_y$. The electrostatic interaction energy between the proton and the $C_xN_y$ with photo-generated electrons is 1.48–4.04 eV, which could readily overcome the proton transport barrier in GR of $\sim$1.23 eV (ref. 36) (Supplementary Table 5). This process is also verified by our *ab initio* MD simulations for proton transfer through the GR sheet in the GR-$C_3N_4$ structure (Supplementary Movie 1, Supplementary Fig. 15), in which parts of photo-generated electrons could be collected by nitrogens. As shown in Fig. 4a, the proton would be chemically bonded with nitrogen. Bader charge analysis found that such N–H bond extracts $\sim$10% of one photo-generated e$^-$ in the whole GR–$C_3N_4$–GR unit. The negative charge can attract another approaching proton, which consequently bonds to the previous H to generate a $H_2$ molecule. The g-$C_xN_y$ desorbs $H_2$ (Fig. 4b)

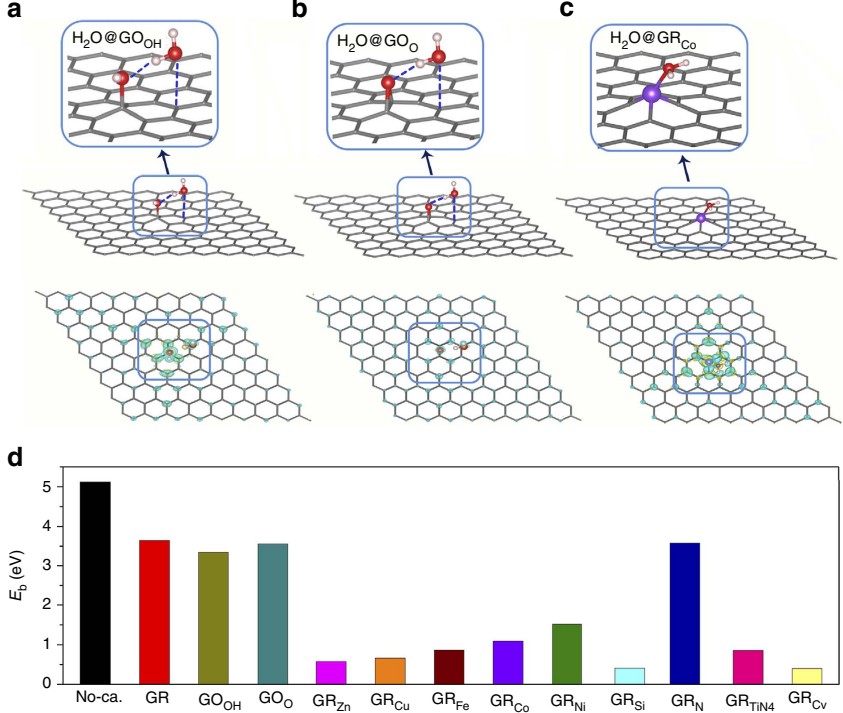

**Figure 3 | Water splitting and proton generation at the outer GR$_F$ surface.** (**a**–**c**) Optimized H$_2$O adsorption geometries and charge distributions (with one photo-generated h$^+$) on GO$_{OH}$, GO$_O$ and GR$_{Co}$ sheets. (**d**) The computed energy barrier ($E_b$) of water splitting reaction catalysed by GR or GR$_F$. Here data of pure GR are retrieved from ref. 30, and No-ca. stands for bare water reaction without catalyst. GO$_{OH}$ and GO$_O$ represent the hydroxyl and epoxy GO; GR$_{Zn}$, GR$_{Cu}$, GR$_{Ee}$, GR$_{Co}$, GR$_{Ni}$ represent the metal-doped GR with Zn, Cu, Fe, Co, Ni atom, respectively; GR$_{Si}$, GR$_N$ stand for Si, N atom-doped GR; GR$_{TiN4}$ represents TiN$_4$-doped GR and GR$_{Cv}$ means for GR with a carbon defect. Yellow and blue bubbles represent electron and hole charges with isosurface value of 0.0005 e Å$^{-3}$. The observation windows in **a**–**c** highlight the local adsorption sites for water molecule.

exothermically, with 0.17 eV for neutral structure while 0.25 eV for negative charged one. Geim and colleagues[36] have demonstrated that GR network allows only proton to pass through. Therefore, O$_2$ and hydroxyl radicals are kept outside from the inside H$_2$ and protons (attracted by negative charges). These thus suppress the unwanted reverse reaction in water splitting, and realize the complete hydrogen evolution process.

**Capsule hydrogen storage.** With sufficient energetic electrons and holes supplies from photo-excited g-C$_x$N$_y$, appreciable amounts of H$_2$ molecules would be produced. As H$_2$ cannot pass through GR-based material, it would be capsuled inside the sandwich. The commercial H$_2$ storage standard postulated by US Department of Energy is 65 kg m$^{-3}$ (hydrogen storage system per se) and 6.5 wt% (by weight)[38]. Although pressurization techniques, container ameliorations and extra absorbents could help to reach this goal, large-scale storage and transportation were limited due to high cost, security risk and technique difficulty. While our sandwich structure can safely capsule H$_2$. We have tested the accommodations of different amounts of H$_2$ in the GR–C$_3$N$_4$–GR sandwich (Fig. 4c, Supplementary Fig. 16), by allowing whole structure relaxation. The interfacial distance increases from 3.0 to 5.5 Å with the store rate from zero to 2.5 wt%, and saturates at 5.5–5.9 Å with store rate from 2.5 to 5.2 wt% (Supplementary Fig. 17). The storage causes almost no structure deformations, and requires nearly negligible energy with 0.001–0.012 eV for 0.2–5.23 wt% store rate (Fig. 4d). In the distance range of 3.6–5.9 Å, effective interlayer couplings still exist and thereby enable ultraviolet–vis light-absorption and hole charge transfer (Supplementary Fig. 18 and Supplementary

Table 6). These agree with previous experimental reports on effective charge transfer between 2D material with interlayer distance of 6–7 Å (refs 25,27). Moreover, the use of pressurization can easily maintain high H$_2$ store rate by controlling interfacial distance. We have found that the interfacial distance can be maintained at ∼3.1 Å by imposing a certain amount of pressure to the outer GR sheet (Fig. 4e). For example, the storage rate of 5.2 wt% can be achieved with ∼3.1 Å interfacial distance by adding ∼56 bar (1 bar = 10$^5$ Pa ≈ 1 atm) external pressure, which is comparable with conditions for other storage systems reported in literature (Supplementary Fig. 19). The rate of 5.2 wt% is close to the standard (6.5 wt%) proposed by US DOE. Although our storage rate is yet to reach the best reported values[39], our design holds unique advantages, such as low cost, and the combination of both hydrogen production and generation.

**Discussion**

In summary, we designed the sandwich structure of one g-C$_x$N$_y$ in between two GR$_F$, to achieve efficient solar harvest, charge separation and reverse reaction inhibition for photocatalytic water splitting, and more importantly to integrate simultaneous hydrogen generation and capsule storage in a single system. It harvests visible and ultraviolet light to generate energetic holes and electrons distributing separately on oxidative and reductive sites of GR$_F$ and g-C$_x$N$_y$. After water splitting on GR$_F$ surface, protons penetrate it through to produce H$_2$ inside the sandwich. By allowing only protons to pass through the outer GR layers, it not only suppresses unwanted reverse reaction, but also capsules H$_2$ products to achieve safe storage. Exploring sp$^2$-carbon sheets (GR, fullerene, carbon-nanotube) to capsule many other

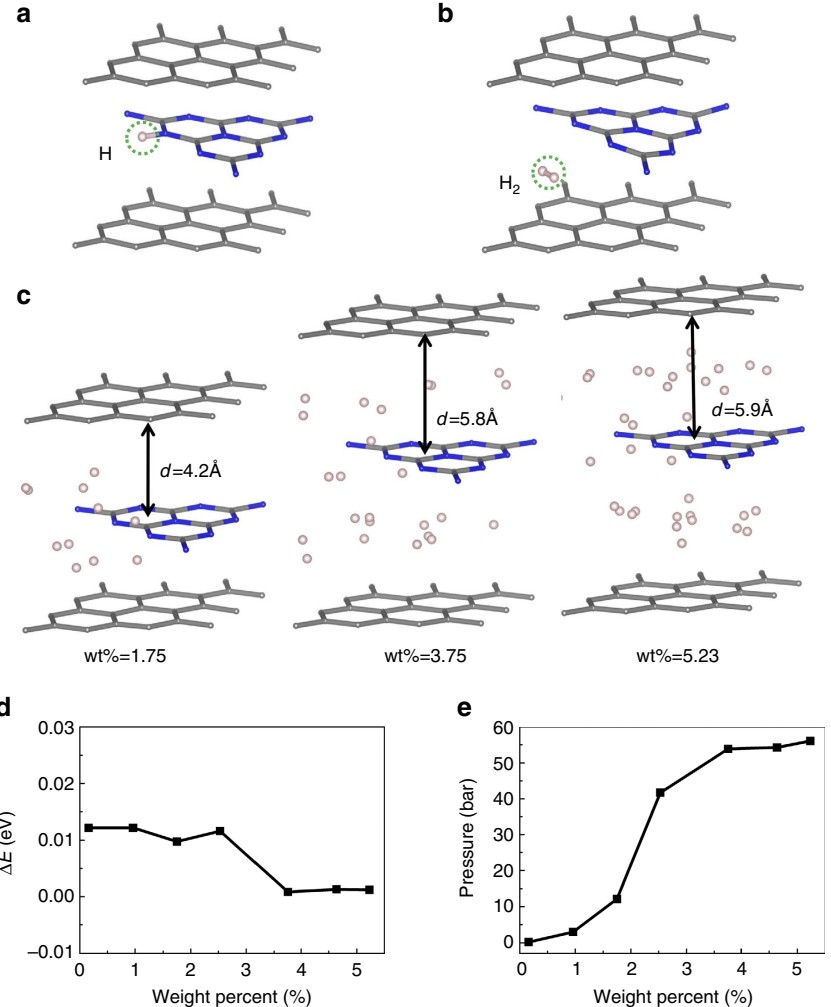

**Figure 4 | Hydrogen evolution and capsule storage.** (**a**–**c**) Optimized configurations of the GR–$C_3N_4$–GR sandwich adsorbed with one H atom, one $H_2$ molecule and many $H_2$ molecules with different interfacial spaces. $d$ in **c** stands for interfacial distance. (**d**) The energy costs ($\Delta E$) to achieve different $H_2$ store rate. (**e**) The variation of the pressure imposed to the outer graphene sheet for GR–$C_3N_4$–GR sandwich structure to achieve different $H_2$ store rate. $1\,\text{bar} = 10^5\,\text{Pa}$ (~1 standard atmospheric pressure).

promising photocatalysts in the same framework would concrete this sandwich concept and provide better candidates for applications. This would stimulate an alternative way of thinking for water splitting towards realistic solar and hydrogen energy utilizations.

## Methods

**Structural optimization and analysis.** All calculations were performed by the Vienna Ab initio Simulation Package at the density functional theory level[40]. Generalized gradient approximation[41], Perdew, Burke, and Ernzerh functional together with projector augmented-wave pseudopotential[42] were employed. State-of-the-art hybrid functional (HSE06) was used for band structure, work function and absorption spectrum calculations[43]. Van der Waals correction was included to account for the nonbonding interaction between GR-based layers and g-$C_xN_y$. The kinetic energy cutoff was set to be 400 eV in the plane-wave expansion. The periodic boundary condition was set with a 15 Å vacuum region above the plane of GR sheet. For the supercell structure, the Morkhost pack mesh of $K$ points was $3 \times 3 \times 1$, while that for single cell was $9 \times 9 \times 1$.

**Transition state calculations.** Climbing image nudged elastic band[44] was used for the search of transition states, and these states were validated by the SSW package HOWTOs program[45].

**Adhesion energy and adsorption energy.** The interface adhesion energy was computed with $E_{ad} = 2E_{GR/GO} + E_{C_xN_y} - E_{sandwich}$, where $E_{GR/GO}$, $E_{C_xN_y}$ and $E_{sandwich}$ represent the energies of the GR/GO, g-$C_xN_y$ and sandwich complex,

respectively. The adsorption of water to $GR_F$ in neutral system was calculated with $E_{ads} = E_{GRF} + E_{H2O} - E_{H2O@GRF}$ ($E_{GRF}$, $E_{H2O}$, $E_{H2O@GRF}$ represent energies of the separated parts and their complex).

**AI-NAMD details.** As for the AI-NAMD calculations[46], we used density functional theory implemented by Vienna Ab initio Simulation Package package to carry out the *ab initio* molecular dynamics. The temperature was set to be 100 K and a 3.5 ps microcanonical trajectory is generated with a time step of 1 fs.

**Data availability.** The authors declare that the data supporting the findings of this study are available within the article and its Supplementary Information files and from the corresponding author upon reasonable request.

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

## Acknowledgements

This work was financially supported by the 973 Program (No. 2014CB848900), NSFC (No. 21633006, 21473166, 21421063, 21373190, 11620101003) and CAS (No. XDB01020000).

## Author contributions

J.J. conceived the research, Y.L. and J.J. supervised the project. L.Y., X.L., X.W. and X.J. performed the simulations. L.Y., X.L., G.Z., P.C., J.Z., Y.L. and J.J. analysed the data. L.Y., X.L., G.Z., Y.L. and J.J. co-wrote the paper. All authors discussed the results and commented on the manuscript.

## Additional information

**Competing interests:** The authors declare no competing financial interests.

