## [Peer Review File · Nature Communications]

Reviewers' comments:

Reviewer #1 (Remarks to the Author):

This manuscript presents a highly creative approach to address the grand challenge that the mixture of oxygen and hydrogen products largely limits the efficiency and safety of photocatalytic water splitting toward practical applications. More importantly, this piece of nice work provides a different angle for cost-effective safe storage of hydrogen molecules -- another bottleneck problem has long hampered the utilization of hydrogen energy. The authors proposed a sandwich design, in which protons from water splitting automatically enter the sandwich through the outer graphene to produce hydrogen molecules, which in turn cannot transport through the graphene and therefore been safely capsule stored. This could be called as a smart material, since such sequential procedures would otherwise require extremely delicate (if not impossible) control on the working mechanism.

The concept is well proven by sophisticated simulations which demonstrate high efficiency for four key steps of photo-absorption, charge separation, hydrogen generation and capsule storage. Hetero-structure designs based on two-dimensional materials are now very promising and already attracted lots of attentions. This work thus clearly advances our understanding and capability in developing two-dimensional materials based catalysts for a variety of applications involving hydrogenation and hydrogen evolution. In addition to catalysis and materials chemistry, this finding also would attract extensive attention from the research community of opto-electric device, organic synthesis and surface science. Overall, this paper is very significant and timely. Thus I would like to recommend this manuscript for publication in Nature communication after a minor revision according to the comments below.

1. It is nature that protons would be driven by negative charges to meet the inner CxNy layer. Is it possible that the interactions between hydrogen and CxNy would affect its light-harvesting ability? I suggest that the authors briefly justify this possibility.
2. I suppose that the storage of hydrogen would enlarge the interfacial distance of the sandwich structure. Would it cause difference to the charge separation process between the outer graphene and inner CxNy layers?
3. The structures of H₂O adsorption on GOOH and GOO in Figure 3a, b are not clear enough. It would be better to provide a big observation window at the local adsorption sites. Meanwhile, charge distribution in Figure 3c, d can also be improved.
4. There are several recent nice publications using two-dimensional hetero-structure materials to achieve good physics or chemistry properties, suggesting the feasibility of experimental synthesis and fabrications. This theoretical work should cite several of those works to show the timely significance of the design.
5. The energy band structure of the materials are missing. I suggest that the authors carry out further simulations to demonstrate the change of electronic structures due to graphene and CxNy interactions.
6. The proposed hydrogen storage rate is quite promising, which is close to the storage standard postulated by US DOE. Nevertheless, it need to be compared with the rate offered by other groups or even the industrial world.
7. Fig 4 presents the equilibrium interfacial distance and energy costs to achieve different store rate. But only the optimized structures of three rates were given in Fig 4c. The authors should do calculations and display the structures of other storage rates, to provide a complete understanding.

Reviewer #2 (Remarks to the Author):

I suggest that the present paper is not suitable for publication in this journal. Ignoring many typos and grammatical issues, only focusing on the underlying science, there are two major reasons along with several less major (still important) issues that led me to this conclusion. These are listed below:

Major issues:

1) Issue with assumption of efficient hole transfer from CxNy to GO. The authors of the manuscript propose a sandwich system of GO-CxNy-GO. The assumption is that the photogenerated exciton is generated and separated in CxNy, after which the electrons stay at CxNy while the holes diffuse to GO (to derive $4h^+ + 2H_2O \rightarrow O_2 + 4H^+$) reaction. This is a bold statement that has not been supported by enough computations and is probably not realistic in this system. Of course the authors try adding a hole or electron to the whole system and observe a hole localization on GO while an electron localization on CxNy. But they do not mention anything about the kinetics and the rates involved in this process which are most probably going to be very low due to the large CxNy-GO distance and the lack of any significant orbital overlap between the two material. Simply showing the most energetically favorable regions for hole and electron localization does not say anything about the rates involved in the actual transfer process. This is a serious issue since in the case that this rate is lower than electron-hole recombination rate on CxNy (which in absence of any internal or external electric field is a high rate) virtually no hole is going to be transferred to the GO sheet. As shown in Fig. 4(c) the distance between GO and CxNy varies in the range of 4.2-5.9 Å, which is only at zero temperature, in any realistic temperature and pressure this distance may increase, which will exponentially lower any rate of hole transfer to the GO sheet hindering any significant H₂ production and storage.

2) Issue with assumption of efficient proton transfer from GO to CxNy. Even ignoring the issue above and simply accepting the assumption of efficient hole transfer to GO, and assuming H⁺ can be formed on GO: what is the rate of H⁺ transfer to CxNy, taking into account that there is at least ~5 Å of vacuum between the GO and CxNy sheet? The manuscript lacks any calculations that even estimate such a rate, let alone a direct calculation.

Minor (still important) issues:

1) In line 60 it is mentioned that always in carbon nitrides electrons are collected at nitrogen positions, this is in direct contradiction with the literature of this subject. For example look at page 36 of the review article (photochemistry reviews) by Dong et al. (<http://dx.doi.org/10.1016/j.jphotochemrev.2014.04.002>) also page 2 of the paper (Nature Materials) by Wang et al. (doi:10.1038/nmat2317) .

2) Based on the manuscript and also Fig 4(c) there is not any significant binding between H₂ and the sheets, which means H₂ molecules only exist in a gaseous state (free to move around and not interacting with the two sandwiching sheets) this translates into this: increasing the concentration of H₂ leads to a linear increase in H₂ gas pressure, which is in direct contrast with the text and self-claimed goal of the manuscript in page 7 "...our metal-free sandwich can safely capsule H₂ without pressurization...".

3) In lines 114-121, the authors mention that H₂O binds very weakly to the GO surface (~0.4 eV, which taking into account entropic term in the Gibbs free energy means that at any relevant temperature or pressure H₂O does not bind to the surface), after this they claim one can increase this binding by more than 4 eV by adding holes to the GO sheet. This sounds completely.

4) In line 124-125 authors mention that the H₂O activation barrier even after lowering due to hole injection becomes 4.38 eV. This is an extremely large barrier that in any realistic temperature translates into virtually zero turnover frequency and zero efficiency.

5) In figure 3 (a) and 3 (b) the O and OH group are bound to a carbon that is part of a 6-carbon rings on graphene, there is no such binding in graphene oxides! This is quite concerning as the computations based on these configurations seem to be a vital part of the manuscript.

Reviewer #3 (Remarks to the Author):

This is an interesting work, bringing out the concept of simultaneous generation and storage of H₂ using a sandwich-type GO-CxNy-GO photocatalyst. The authors need to address the following questions before this paper can be considered further.

1. Why are the photo-induced holes in CxNy injected into the GO layers and the induced electrons retained in CxNy? The authors should provide the physical meaning and significance underlining this vectorial charge transfer process, not just providing the calculation results.
2. The H₂ storage capacity must be low, otherwise the generated H₂ would destroy the sandwich structure. Can the authors estimate the capacity for H₂ storage in this sandwich system?
3. The authors should provide an experimental trial to prove that the combination of graphene and CxNy is capable of simultaneously generating H₂ and O₂ under light irradiation.

Responses to the reports of Reviewers

Responses to the Reviewer 1

General comment: *This manuscript presents a highly creative approach to address the grand challenge that the mixture of oxygen and hydrogen products largely limits the efficiency and safety of photocatalytic water splitting toward practical applications. More importantly, this piece of nice work provides a different angle for cost-effective safe storage of hydrogen molecules – another bottleneck problem has long hampered the utilization of hydrogen energy. The authors proposed a sandwich design, in which protons from water splitting automatically enter the sandwich through the outer graphene to produce hydrogen molecules, which in turn cannot transport through the graphene and therefore been safely capsule stored. This could be called as a smart material, since such sequential procedures would otherwise require extremely delicate (if not impossible) control on the working mechanism.*

The concept is well proven by sophisticated simulations which demonstrate high efficiency for four key steps of photo-absorption, charge separation, hydrogen generation and capsule storage. Hetero-structure designs based on two-dimensional materials are now very promising and already attracted lots of attentions. This work thus clearly advances our understanding and capability in developing two-dimensional materials based catalysts for a variety of applications involving hydrogenation and hydrogen evolution. In addition to catalysis and materials chemistry, this finding also would attract extensive attention from the research community of opto-electric device, organic synthesis and surface science. Overall, this paper is very significant and timely.

Thus I would like to recommend this manuscript for publication in Nature communication after a minor revision according to the comments below.

Response: We are very grateful to the reviewer for his/her appreciation of our findings and his/her most positive report. In the following, we provide concrete responses to the comments and suggestions from the reviewer point-by-point.

Comment 1: *It is nature that protons would be driven by negative charges to meet the inner C_xN_y layer. Is it possible that the interactions between hydrogen and C_xN_y would affect its light-harvesting ability? I suggest that the authors briefly justify this possibility.*

Response: We thank the reviewer for this important suggestion. We have examined the photo-absorption ability of Gr- C_3N_4 -Gr sandwiched structure with the hydrogen storage rate at 0.96 wt% (interlayer distance $d=3.6$ Å), 1.75wt% (interlayer distance 4.2 Å), and 5.23wt% (interlayer distance 5.9 Å), respectively. It is found that these structures are still able to harvest both visible and ultraviolet photons. We have discussed these results in Line 15-18/Paragraph 2/Page 9 in the revised manuscript and Supplementary Fig. 12.

Comment 2: *I suppose that the storage of hydrogen would enlarge the interfacial distance of the sandwich structure. Would it cause difference to the charge separation process between the outer graphene and inner C_xN_y layers?*

Response: The reviewer is right that the storage of hydrogen can enlarge the interfacial distance of the sandwich structure. We have examined the effect of

interfacial distance on the charge separation by calculating the charge distribution of the sandwiched Gr-C₃N₄-Gr at 3.1 Å, 3.6, 4.2, and 5.9 Å, respectively. At the interfacial distance of 3.1 Å, one photo-generated electron carrier could induce about 0.27 e⁻ in the C₃N₄ sheet and 0.97 h⁺ in the Gr cells. As for the interfacial distance of 3.6, 4.2, 5.9 Å, those numbers become slightly smaller, (~0.26 e⁻ and 0.92 h⁺), (~0.16 e⁻ and 0.88 h⁺), (~0.16 e⁻ and 0.70 h⁺), respectively. It is clear that the enlarged interfacial distance can maintain effective charge separation. It also agrees well with previous experimental reports on effective charge transfer between 2D materials with interlayer distance of 6~7 Å (Nat. Nanotechnol., 2014, 9, 682; J. Am. Chem. Soc., 2015, 137, 8313. Ref. 24 and 26, respectively, in the revised version).

It is also worth to mention that by adding external pressure, the interfacial distance can be quickly reduced to the original value by adding <56 bar external pressure (Fig. 4e), which is comparable with the conditions and rates in previous reports (Supplementary Fig. 14).

We have added the discussions in Line 15-22/Paragraph 2/Page 9 and Line 1-5/Paragraph 1/Page 10 of the revised version, as well as the Fig. 4e and Supplementary Table 4 and Fig. 14.

Comment 3: *The structures of H₂O adsorption on GO_{OH} and GO_O in Figure 3a, b are not clear enough. It would be better to provide a big observation window at the local adsorption sites. Meanwhile, charge distribution in Figure 3c, d can also be improved.*

Response: We thank the reviewer for pointing out this problem. We have changed the figure to include big observation windows at the local adsorption sites for Figure 3a-d to improve the presentation.

Comment 4: *There are several recent nice publications using two-dimensional hetero-structure materials to achieve good physics or chemistry properties, suggesting the feasibility of experimental synthesis and fabrications. This theoretical work should cite several of those works to show the timely significance of the design.*

Response: We thank the reviewer for the good suggestion. We have now cited several recent works on 2D hetero-structure materials in the references (Ref. 19. Nat. Nanotechnol., 2016, 11, 218; Ref. 20. Nature Commun., 2016, 7, 13911; Ref. 21. Nature Commun., 2016, 7, 12697), which indeed support the synthesis and fabrication feasibilities of our designs.

Comment 5: *The energy band structure of the materials are missing. I suggest that the authors carry out further simulations to demonstrate the change of electronic structures due to graphene and C_xN_y interactions.*

Response: We thank the reviewer for bringing this into our attention. In the revised manuscript, we have given the band structures of the materials calculated at the HSE06 level in Supplementary Fig. 3. The calculated energy band structures of the bare CN, C_2N , C_3N_4 agree well with previously reported results. (Chem. Commun., 2016, 52, 13233(see Ref. 29); Phys. Chem. Chem. Phys., 2016, 18, 3144; J. Phys. Chem. C, 2014, 118, 26479). And the sandwiched Gr- C_xN_y -Gr structures all lead to the band gap opening in the graphene due to

the interaction of the graphene and C_xN_y (J. Am. Chem. Soc., 2012, 134, 4393(see Ref. 14); RSC Adv., 2016, 6, 28484; J. Phys. Chem. C, 2015, 119, 28417). We also added the discussion in Line 2-4/Paragraph 1/Page 6 of the revised manuscript.

Comment 6: *The proposed hydrogen storage rate is quite promising, which is close to the storage standard postulated by US DOE. Nevertheless, it need to be compared with the rate offered by other groups or even the industrial world.*

Response: We agree with the reviewer that the promising property of our designed materials should be put together with other existing results. In the revised manuscript, the variation of the pressure as a function of the hydrogen storage rates for previously reported systems is displayed (Supplementary Fig. 14). It can be seen that in contrast to previous storage recipes that often rely on external pressure, our non-pressurized sandwiched structure could already achieve a relatively high storage rate. We have further found that our materials can result in the storage rate of 5.2 wt% with interfacial distance ~ 3.1 Å by adding ~ 56 bar external pressure, which is comparable with the previous reports (Supplementary Fig. 14). This projected hydrogen storage rate is close to the storage standard (6.5 wt%) postulated by US DOE. Although the storage rate of our proposed system is yet to reach the best reported values offered by other materials such as MOFs, COFs, and hydrides of light elements, (Ref. 34. Nat. Rev. Mater., 2016, 1, 16059), if considering the cost of the materials, the advantage of our design is quite obvious. Moreover, our sandwiched structure can effectively integrate photocatalytic hydrogen generation and safe capsule storage.

These viewpoints are discussed in Line 19-22/Paragraph 2/Page 9, Line 1-5/Paragraph 1/Page 10 and Fig. 4e in the revised manuscript, as well as in Supplementary Fig. 14 of the revised Supporting Information.

Comment 7: *Fig 4 presents the equilibrium interfacial distance and energy costs to achieve different store rate. But only the optimized structures of three rates were given in Fig 4c. The authors should do calculations and display the structures of other storage rates, to provide a complete understanding.*

Response: We are grateful to the reviewer for this important suggestion. The structures of the other three storage rates (0.96 wt%, 2.53 wt%, 4.64 wt%) are collected in Supplementary Fig. 11 of the revised Supporting Information.

Responses to the Reviewer 2

General comment: *I suggest that the present paper is not suitable for publication in this journal. Ignoring many typos and grammatical issues, only focusing on the underlying science, there are two major reasons along with several less major (still important) issues that led me to this conclusion. These are listed below:*

Response: We should have done a better job to avoid typos and grammatical issues in the original manuscript. We have carefully checked the manuscript and made all necessary corrections. Our point-by-point responses to the report of the reviewer are given below.

Major issues:

Comment 1: *Issue with assumption of efficient hole transfer from C_xN_y to GO. The authors of the manuscript propose a sandwich system of GO- C_xN_y -GO. The assumption is that the photogenerated exciton is generated and separated in C_xN_y , after which the electrons stay at C_xN_y while the holes diffuse to GO (to derive $4h^+ + 2H_2O \rightarrow 2H_2 + 4H^+$) reaction. This is a bold statement that has not been supported by enough computations and is probably not realistic in this system. Of course the authors try adding a hole or electron to the whole system and observe a hole localization on GO while an electron localization on C_xN_y . But they do not mention anything about the kinetics and the rates involved in this process which are most probably going to be very low due to the large C_xN_y -GO distance and the lack of any significant orbital overlap between the two material. Simply showing the most energetically favorable regions for hole and*

electron localization does not say anything about the rates involved in the actual transfer process. This is a serious issue since in the case that this rate is lower than electron-hole recombination rate on C_xN_y (which in absence of any internal or external electric field is a high rate) virtually no hole is going to be transferred to the GO sheet. As shown in Fig. 4(c) the distance between GO and C_xN_y varies in the range of 4.2-5.9 Å, which is only at zero temperature, in any realistic temperature and pressure this distance may increase, which will exponentially lower any rate of hole transfer to the GO sheet hindering any significant H_2 production and storage.

Response: We agree with the reviewer that the electron-hole separation holds the key of the design given in this work. The rate of electron-hole separation is a very different parameter to be determined reliably from either experiment or computation. The reviewer asked for something that cannot be completely resolved from current technologies. However, we would like to emphasize that there are enough experimental and theoretical evidences, together with our new calculations, to support the notion of fast electron-hole separation in the designed stacked systems, as given below:

- (1) **The work function difference between C_xN_y and graphene is the driving force for the charge separation** (Supplementary Fig. 4). Due to the lower work function of Gr/GO, the Gr/GO layers could extract the hole from the C_xN_y sheet. The driving force caused by the work function difference has been demonstrated in many previous theoretical and experimental works (Nature Commun., 2015, 6, 6485(see Ref. 32); J. Am. Chem. Soc., 2014, 136, 14650; Nano Res., 2015, 8, 3621; J. Mater. Chem. C, 2015, 3, 5089; Nano Lett., 2015, 15, 6475).
- (2) **The strong interlayer electron/orbital couplings between the stacked C_xN_y and graphene layers provide channels for the charge transfer.**

Recent works (J. Am. Chem. Soc., 2012, 134, 4393. See Ref.14; RSC Adv., 2016, 6, 28484; J. Phys. Chem. C, 2015, 119, 28417) have shown that the strong orbital coupling between g-C₃N₄ and graphene is responsible for the gap opening in the g-C₃N₄-supported graphene, as well as the enhancement of the electron conductivity and the expanded absorption spectrum from UV into the visible region. Our results are consistent with theirs, the Gr/GO–C_xN_y–Gr/GO sandwiched systems also displayed the gap opening of graphene, the broadening of the absorption spectrum, and the electron–hole exchanges at the hybrid structures (Fig. 2, Supplementary Fig. 2-3, 5, 7-8, 15-16).

- (3) **The fast dynamics of the charge transfer in ps time scale favors the charge separation.** Encouraged by this comment, we have carried out time dependent ab initio non-adiabatic molecular dynamics (AIMD) simulations to elucidate the fundamental aspects of the ultrafast hole dynamics. The simulation result for the charge transfer dynamics is shown in Supplementary Fig. 6 of the revised manuscript. It is found that the hole carrier with lower energy (near the valence band) can transfer from CN to Gr, and ~14% of the carrier would reach the Gr sheet within 3 ps. For the hole with higher energy, more than half of the carriers would transfer within 0.4 ps, and ultimately, above 80% carriers would be transferred from the CN to Gr layer. It can be concluded that in our hybrid structure, the photo-generated hole can transfer from C_xN_y to Gr sheet within a few ps. This relatively fast dynamics makes subsequent reactions (i.e. water splitting on GO, and H₂ generation on C_xN_y) feasible.
- (4) **Reliable experimental results clearly support the ultrafast charge transfer within van der Waals 2D layers.** Recent experiments have disclosed efficient electron-hole separation in vertically stacked transition-metal dichalcogenide (TMD) heterostructure bilayers, through ultrafast charge transfer between two adjacent layers with interlayer distance 6 ~ 7 Å

and vdW interactions (Nat. Nanotechnol., 2014, 9, 682(see Ref. 24); Nat. Nanotechnol., 2014, 9, 676(see Ref. 25); ACS Nano, 2014, 8, 12717; J. Am. Chem. Soc., 2015, 137, 8313(see Ref. 26)). The shortest timescale was obtained in ultrafast measurements on MoS₂/WS₂ heterostructures by Hong et al (Nat. Nanotechnol., 2014, 9, 682, Ref. 24) with an upper limit of 50 fs.

- (5) **C_xN_y is known to be able to maintain charge separations for effective hydrogen evolutions.** Kang et al (Science, 2015, 347, 970, see Ref. 12) recently reported a nanocomposite of carbon nanodot–C₃N₄ with stable visible-driving water splitting, where the transfer of photo-generated holes from C₃N₄ to carbon nanodot enables the water splitting for O₂ generation. Wang et al (Angew. Chem. Int. Ed., 2016, 128, 1) demonstrated photo-generated electrons and holes can transfer from C₃N₄ to the contacted Pt and Co₃O₄ nanoparticles, separately, substantially suppressing charge recombination.
- (6) **The interfacial distance is within the right range to provide effective charge separation.** The optimized interfacial distance is 2.94~3.26 Å for all of the systems under investigations, which ensures effective transferring of the photo-generated carriers (Supplementary Table 1). We have tested the systems with enlarged distance of 3.6, 4.2 and 5.9 Å. The narrowing of band gap due to strong couplings still exist (Supplementary Fig. 12), and the charge separation remains (Supplementary Table 4). These agree with previous experimental reports of effective charge transferring between 2D materials with interlayer distance of 6~7 Å. (Nat. Nanotechnol., 2014, 9, 682(see Ref. 24); J. Am. Chem. Soc., 2015, 137, 8313(see Ref. 26)).
- (7) **External pressure can be added to maintain best interlayer contacts for effective light harvesting and charge transferring.** For instance, to maintain the interfacial distance at ~3.1 Å with an increased H₂ storage, a growing pressure in a reasonable range can be imposed to the outer graphene sheet accordingly (Fig. 4e). A storage rate of 5.2 wt% is achieved with ~3.1

Å interfacial distance by adding ~56 bar, which is comparable with conditions for other storage systems in literature (Supplementary Fig. 14). In this case, the effective separation of electrons and holes for the photocatalytic reaction can still be achieved.

We have added the above discussions in Line 2-6/Paragraph 1/Page 4, Line 2-4/Paragraph 1/Page 6, Line 8-11, 15-22/Paragraph 2/Page 6-7, Line 14-22/Paragraph 2/Page 9 and Line 1-5/Paragraph 1/Page 10, Fig. 2, Fig. 4e, Table 1 in the revised manuscript and Supplementary Fig. 2-8, 12, 14-16, Table 1, 4 in the Supporting Information.

Comment 2: *Issue with assumption of efficient proton transfer from GO to C_xN_y. Even ignoring the issue above and simply accepting the assumption of efficient hole transfer to GO, and assuming H⁺ can be formed on GO: what is the rate of H⁺ transfer to C_xN_y, taking into account that there is at least ~5 Å of vacuum between the GO and C_xN_y sheet? The manuscript lacks any calculations that even estimate such a rate, let alone a direct calculation.*

Response: We thank the reviewer for establishing the common ground for the further discussion by accepting the assumption of efficient hole transfer to GO. Here we discuss the issue of the proton transport from different point of views.

The proton transport barrier through graphene is already known to be ~1.23 eV (Nature, 2014, 516, 227, see Ref. 31). Geim et al. verified that the energy barrier could be overcome by electrostatic attractions between protons and negative charged metals. Recently, Zhang et al (Nano Lett., 2016, 16, 6058) utilized this penetrability to enhance the hydrogen activation reactivity of nonprecious metal substrates by the confined underneath graphene sheet. These all confirm the feasibility of the proton penetration through graphene.

For our sandwiched systems, the proton migration can be driven by the electrostatic attractions between protons and the C_xN_y layer with photo-induced negative charges. We have carried out ab initio MD simulations to verify the proton transfer process through the graphene sheet in the Gr- C_3N_4 structure (Supplementary Fig. 10). To describe the rate quantitatively, we have now calculated the electrostatic interaction energy between the proton and the C_xN_y with photo-generated electrons in the sandwiched structures (Supplementary Table 3). The vacuum between the GO and C_xN_y in the original systems is within 2.94~3.26 Å, for which the smallest coulomb interaction energy is found to be 1.48 eV, much larger than the proton penetration barrier of 1.23 eV.

Even for the interfacial distance of ~5.0 Å after hydrogen storage, the coulomb interaction energy is still be of 0.92~2.51 eV, which is feasible to overcome the proton penetration barrier. Moreover, the interfacial distance can be reduced by applying external pressure, which will guarantee the efficiency of the proton transport as discussed in the reply above and Fig. 4e.

The new discussion is given in Line 10-14/Paragraph 2/Page 8, Line 19-22/Paragraph 2/Page 9 and Line 1-5/Paragraph 1/Page 10, Fig. 4e of the revised manuscript together with Supplementary Fig. 10, Table 3 of the revised Supporting Information.

Minor (still important) issues:

Comment 1: *In line 60 it is mentioned that always in carbon nitrides electrons are collected at nitrogen positions, this is in direct contradiction with the literature of this subject. For example look at page 36 of the review article*

(photochemistry reviews) by Dong et al. (<http://dx.doi.org/10.1016/j.jphotochemrev.2014.04.002>) also page 2 of the paper (Nature Materials) by Wang et al. (doi:10.1038/nmat2317).

Response: We thank the reviewer for pointing out this presentation problem. The reviewer is right that the statement in line 60 could lead to misunderstanding. Our calculated results (Fig. 2, Supplementary Fig. 5, 7, 8) also show clearly that electrons are not only located at the nitrogen positions, but also presented around carbons in the C_xN_y sheet. Our findings are in line with previous literatures by Dong et al and Wang et al. To make it clearer, we have changed the sentence from “Our theoretical and experimental work demonstrate that carbon nitrides always collect electrons at nitrogen positions” to “Our theoretical and experimental works demonstrate that parts of photo-generated electrons in carbon nitrides are collected at nitrogen positions”, in the Line 7-9/Paragraph 1/Page 4 of the revised manuscript.

Comment 2: *Based on the manuscript and also Fig 4(c) there is not any significant binding between H_2 and the sheets, which means H_2 molecules only exist in a gaseous state (free to move around and not interacting with the two sandwiching sheets) this translates into this: increasing the concentration of H_2 leads to a linear increase in H_2 gas pressure, which is in direct contrast with the text and self-claimed goal of the manuscript in page 7 “...our metal-free sandwich can safely capsule H_2 without pressurization...”.*

Response: The reviewer is right. As we discussed also in replies above, the pressurization will become a key helper to achieve the goals of the designed systems. We have examined the influence of pressurization on the hydrogen

storage rate. It is found that the interfacial distance can be maintained at ~ 3.1 Å by imposing a certain amount of pressure to the outer graphene sheet (Fig. 4e). For example, the storage rate of 5.2 wt% is achieved with ~ 3.1 Å interfacial distance by adding ~ 56 bar external pressure, which is comparable with the conditions of other H₂ storing systems in previous reports (Supplementary Fig. 14). Such pressures could be achieved experimentally. Moreover, the adding of the pressure helps also to promote the proton permeability and kinetic enhancement.

The related discussion can be found in Line 19-22/Paragraph 2/Page 9, Line 1-5/Paragraph 1/Page 10 and Fig. 4e of the revised manuscript and Supplementary Fig. 14 of the Supporting Information.

Comment 3: *In lines 114-121, the authors mention that H₂O binds very weakly to the GO surface (~ 0.4 eV, which taking into account entropic term in the Gibbs free energy means that at any relevant temperature or pressure H₂O does not bind to the surface), after this they claim one can increase this binding by more than 4 eV by adding holes to the GO sheet. This sounds completely.*

Response: We thank the reviewer for bringing this issue to our attention. It has made us carefully re-examine our model used for computing the binding energy.

In our original calculations, the adsorption energy of water to GO with 1 h⁺ (hole carrier) injection was computed by the equation: $E_a = E_{GO}(h^+) + E_{H_2O} - E_{H_2O@GO}(h^+)$, where $E_{GO}(h^+)$, E_{H_2O} , $E_{H_2O@GO}(h^+)$ represent energies of the isolated GO part with a hole, isolated water molecule, and their complex with a hole, respectively. The energy of $E_{GO}(h^+)$ was used as a “good approximation” since

the majority of the hole is collected by the GO part (eg. ~ 0.97 holes in GO_{OH} for the 1 h^+ injected $\text{H}_2\text{O}@_{\text{GO}_{\text{OH}}}$). Thanks to the reviewer's comment, we realized that such an approximation could result in larger binding energy due to the lack of hole effect on the water.

We have re-calculated that adsorption energy with a more reasonable equation: $E_a = E_{\text{GO}+\text{H}_2\text{O}}(\text{h}^+) - E_{\text{H}_2\text{O}@_{\text{GO}}}(\text{h}^+)$. Therein, $E_{\text{GO}+\text{H}_2\text{O}}(\text{h}^+)$ gives the energy of the total system with one hole, in which GO and H_2O are far from each other without interaction. This model is closer to the actual system. As a result, the computed adsorption energies are increased from 0.37 and 0.32 eV at the neutral states to 0.93 and 0.94 eV for GO_{OH} and GO_0 with one hole, respectively. This gives much more reasonable increasing of the adsorption energy due to the presence of the extra hole (refer to new Fig. 3e).

The new results are given in Line 16/Paragraph 2/Page 7 and Fig. 3e of the revised manuscript.

Comment 4: *In line 124-125 authors mention that the H_2O activation barrier even after lowering due to hole injection becomes 4.38 eV. This is an extremely large barrier that in any realistic temperature translates into virtually zero turnover frequency and zero efficiency.*

Response: The reviewer is right about the value and its consequence. It should be emphasized that this value corresponds to the energy barrier for breaking O-H bond in a bare H_2O molecule without any catalysts. The well-documented value for this barrier is ~ 5.12 eV (Nelson, D. L.; Cox, M. M.; Lehninger, A. L., *Lehninger principles of biochemistry*. 4th ed.; W.H. Freeman: New York, 2005; p 48.). It can be significantly reduced by ~ 0.75 eV with the help of one hole

carrier, as our calculation confirmed. The presence of catalytic GO can significantly reduce this energy barrier to 2.83 eV for the water molecule adsorbed to GO_{OH}, if only ~0.03 effective hole carrier is distributed on the water when the 2h⁺ is injected to the whole system. We expect that in the actual reaction process, appreciable amounts of holes would transfer to water, and the energy barrier of water splitting could be reduced even more.

We should also mention that the difficulty in precisely taking into account the energetic hole in electronic structure calculation makes it only possible to provide an approximate estimation. But the trend of the reduction for the reaction barrier is clearly demonstrated in our calculations, which suggest the catalytic role of GO with the hole for water splitting.

In this context, the experimental evidence is more convincing. It has been reported in the literatures that functional groups in GO and large accessible surface make it an effective medium for water decomposition. Teng et al (Adv. Funct. Mater., 2010, 20, 2255) for the first time demonstrated the GO specimen as a photocatalyst for the steady evolution of H₂ from solar water splitting. Later they also reported several strategies to tune the electronic structure of GO for photocatalytic water splitting (J. Phys. Chem. C, 2011, 115, 22587; Mater. Today, 2013, 16, 3, 78 (see Ref. 27); Adv. Mater., 2014, 26, 3297; Nano Energy, 2015, 12, 476). Therein, with the dopant of N atoms or extending of oxidation degree, GO could become an ideal material for the overall water splitting. Previous literatures (ACS Nano, 2010, 6, 3169; Adv. Mater., 2013, 25, 3820; Adv. Mater., 2015, 25, 872) also suggested the injection of holes into the valence of GO, which could be used for light-driven water splitting. These experimental results demonstrated the feasibility of water splitting reaction for our designed structures.

Last but not least, our Gibbs free energy results demonstrated that hole injection could effectively reduce the Gibbs free energy of this water splitting reaction (0.47 eV and 0.67 eV for GO_{OH} and GO_O respectively, comparing to the neutral one), which is favorable for the water splitting in terms of thermodynamics (Supplementary Table 2).

The related discussion can be found in Line 20-22/Paragraph 2/Page 7, Line 1-8/ Paragraph 1/Page 8 in the revised manuscript and Supplementary Fig. 9, Table 2 of the Supporting Information.

Comment 5: *In figure 3 (a) and 3 (b) the O and OH group are bound to a carbon that is part of a 6-carbon rings on graphene, there is no such binding in graphene oxides! This is quite concerning as the computations based on these configurations seem to be a vital part of the manuscript.*

Response: We can agree that the GO can have a very complicated structure. However, in terms of calculations, our model for GO is the same and reliable as many others reported in previous studies. Just to name a few: the model for the graphene oxide in Nature Commun., 2015, 6, 8335, as depicted in Figure 1 below. The initial configuration of MD simulations on GO sheets performed by Shenoy et al (Nature Chem., 2010, 2, 581) based on the observations of Cai et al (Science, 2008, 321,1815) is shown in Figure 2 below. We can also mention that in articles, such as J. Phys. Chem. C 2015, 119, 18167, Sci. Rep., 2013, 3, 2484, RSC Adv., 2015, 5, 11966, the same configuration in which O and OH groups are bound to a carbon belong to a 6-carbon rings on graphene is adopted.

Figure 1. It is illustrated in the literature that “Graphene oxide (GO) is used as a starting material for the preparation of reduced graphene oxide (rGO). Epoxy and hydroxyl functional groups are randomly distributed on both sides of the starting GO sheet.”(Nature Commun., 2015, 6, 8335).

Figure 2. It is illustrated in the literature that “(a) Initial configuration of hydroxyl and epoxy groups used in the MD calculations based on the observations of Cai et al.”(Nature Chem., 2010, 2, 581).

Responses to the Reviewer 3

General comment: This is an interesting work, bringing out the concept of simultaneous generation and storage of H₂ using a sandwich-type GO–C_xN_y–GO photocatalyst. The authors need to address the following questions before this paper can be considered further.

Response: We are very grateful to the reviewer for his/her appreciation of the new concept introduced in this work. In the following, we provide concrete responses to the comments and suggestions from the reviewer point-by-point.

Comment 1. *Why are the photo-induced holes in C_xN_y injected into the GO layers and the induced electrons retained in C_xN_y ? The authors should provide the physical meaning and significance underlining this vectorial charge transfer process, not just providing the calculation results.*

Response: We thank the reviewer for pointing out the key issue of the whole design. We have carried out some extra calculations to address the comment.

The separation of the electron and the hole in our system is driven by the difference of work functions between C_xN_y and Gr/GO. As shown in Supplementary Fig. 4, the work functions of Gr/GO are much lower than the C_xN_y material (1.7~2.9 eV lower). This could naturally induce the electron flow from the graphene-based sheet (lower work function) to the C_xN_y structure (higher work function). The same mechanism has been verified by several previous theoretical and experimental works (J. Am. Chem. Soc., 2014, 136, 14650; Nature Commun., 2015, 6, 6485 (see Ref. 32); Nano Res., 2015, 8, 3621-3629; J. Mater. Chem. C, 2015, 3, 5089; Nano Lett., 2015, 15, 6475). All these results suggest that the photo-induced electrons in graphene could be successfully transferred to the C_xN_y materials.

We have carried out time dependent ab initio non-adiabatic molecular dynamics (AI-NAMD) simulations to describe the ultrafast hole dynamics. It can directly confirm the hole transfer process from CN sheet to graphene layers. In Supplementary Fig. 6, it can be seen that the hole with lower energy (near the

valence band) could transfer from CN to Gr, and ~14% hole carriers would reach to the Gr sheet within 3 ps. For the hole with higher energy, more than half of the hole carriers would be quickly transferred within 0.4 ps. Ultimately, above 80% hole carriers in CN layer would transfer to the Gr layer. It can be concluded that in our hybrid structure, the photo-generated hole transfers from C_xN_y to Gr sheet within ps.

The related discussion can be found in Line 8-11, 15-22/Paragraph 2/Page 6-7 in the revised manuscript and Supplementary Fig. 4, Fig. 6 of the Supporting Information.

Comment 2: *The H_2 storage capacity must be low, otherwise the generated H_2 would destroy the sandwich structure. Can the authors estimate the capacity for H_2 storage in this sandwich system?*

Response: In our sandwiched system, the generated hydrogen could cause minor structure deformation when the storage rate changes from 0.2 to 5.2 wt%, if we allow the whole structure to relax without adding any extra pressurization (Fig. 4e). We could say that the capacity for H_2 storage in this sandwich system is quite impressive.

Obviously, the adding of external pressure will further enhance its capacity. New calculations have shown that the interfacial distance can be maintained at ~3.1 Å by imposing a certain amount of pressure to the outer graphene sheet (Supplementary Fig. 13). For example, the storage rate of 5.2 wt% could be achieved with the interfacial distance of ~3.1 Å by adding ~56 bar external pressure, which is comparable with the conditions for other H_2 storing systems reported in previous studies (Supplementary Fig. 14). Such pressures could be

achieved experimentally. Moreover, the adding of the pressure helps also to promote the charge separation, the proton permeability and kinetic enhancement.

The related discussion can be found in Line 19-22/Paragraph 2/Page 9 and Line 1-5/Paragraph 1/Page 10, Fig. 4e of the revised manuscript and Supplementary Fig. 13, Fig. 14 of the Supporting Information.

Comment 3: *The authors should provide an experimental trial to prove that the combination of graphene and C_xN_y is capable of simultaneously generating H_2 and O_2 under light irradiation.*

Response: We thank the reviewer for his/her truth on our new concept and the designed systems. We are of course eager to quickly turn our concepts into the reality. Unfortunately, for a group of theorists like us, this task becomes a “mission impossible”.

However, we are very confident that once the concepts get published, many excellent experimental groups will quickly follow the idea to fabricate these designed systems and to explore their potentials. As we all know, there are synthetic and fabrication methods available to make such sandwiched structures.

Reviewers' comments:

Reviewer #1 (Remarks to the Author):

Now, I think this one should be published in Nat.Commun.

Reviewer #2 (Remarks to the Author):

The authors of the manuscript have included explanations, corrections and supplementary information to the manuscript that has definitely improved the quality of the manuscript.

Nevertheless, there is still some ambiguity on the feasibility of their scheme based on the number they report for the water dissociation process in page 14 of their response ~ 2.83 . This is simply too large of a barrier for effective water splitting at room temperature. It might be that they need another functional group or active site embedded in graphene network to catalyze and further decrease this energy barrier, which is not currently considered in this scheme. A barrier of 2.83 eV leads to virtually zero TOF at room temperature, thus renders their method ineffective.

I believe this is a serious point that the authors have to address before their manuscript could be considered further for publication.

Reviewer #3 (Remarks to the Author):

Simultaneous evolution of H₂ and O₂ from photocatalysis remained a challenge in the field of photo-driven water-splitting. Theoretical calculation can only act as a supporting role in this research area, and a preliminary experimental study, at least, is essential to justify interpretations derived from theoretical calculations. If the authors' calculations are practically applicable, all the photocatalyst nanoparticles, with suitable band positions, dispersed between two GO layers can provide the same functions as those given by C₃N₄. The reviewer strongly suggests the authors to ask supports from experimentalists for justification of their calculations.

Responses to the reports of Reviewers

Responses to the report of the Reviewer 1

General comment: Now, I think this one should be published in Nat. Commun.

Response: We are very grateful to the reviewer.

Responses to the report of the Reviewer 2

General comment: The authors of the manuscript have included explanations, corrections and supplementary information to the manuscript that has definitely improved the quality of the manuscript.

Response: We are very grateful to the reviewer for his/her appreciation on our efforts.

Comment 1. Nevertheless, there is still some ambiguity on the feasibility of their scheme based on the number they report for the water dissociation process in page 14 of their response ~ 2.83. This is simply too large of a barrier for effective water splitting at room temperature. It might be that they need another functional group or active site embedded in graphene network to catalyze and further decrease this energy barrier, which is not currently considered in this scheme. A barrier of 2.83 eV leads to virtually zero TOF at room temperature, thus renders their method ineffective.

I believe this is a serious point that the authors have to address before their

manuscript could be considered further for publication.

Response: We thank the reviewer for the insightful comment. We agree that the consideration of other functional groups could be the solution to increase the water splitting efficiency.

We have added a couple of functional groups and active sites that are known to have good catalysis performance for water splitting (Ref. 19. Nat. Nanotechnol., 2016, 11, 218; Ref. 28. Angew. Chem. Int. Ed., 2015, 127, 14237; Ref. 29. Sci. Rep., 2014, 4, 6450; Ref. 31. Chem. Soc. Rev., 2009, 38, 253) on our graphene systems. The new functionalized sandwich system is now labeled as $\text{GR}_F\text{-C}_x\text{N}_y\text{-GR}_F$ system, where GR_F means graphene modified by functional groups. By carrying out the Nudged Elastic Band (NEB) calculations, we found that the energy barrier of water splitting reaction (E_b) can be significantly reduced from 5.13 eV (bare H_2O molecule) to ~ 0.5 eV with certain GR_F systems. As listed in Fig 3d, Supplementary Fig. 9-13, and Supplementary Table 2, the E_b value is 3.34~3.56 for GR with oxygen functional groups (GO_{OH} and GO_{O}), 0.58~1.09 eV for GR with metal atom (Metal: Zn, Cu, Fe, Co), 0.41 eV for GR with Si atom, 0.86 eV for GR with the TiN_4 group, and 0.40 eV for GR with a carbon vacancy defect (C_v). Meanwhile, the adsorption of water molecule to these functional groups is also found to be feasible with relatively high adsorption energies as listed in Supplementary Table 2. Most importantly, the separation of photo-induced charges in these sandwich systems remains to be efficient with the outer GR_F layers collecting hole carriers (Supplementary Fig. 14 and Table 3). The much-reduced water splitting energy barriers should thus enable to generate efficient photocatalytic reaction in our proposed scheme.

These viewpoints are discussed in Line 3-15/Paragraph 2/Page 7 and Line 1-4/Paragraph 1/Page 8, Fig. 3d in the revised manuscript, as well as in Supplementary Fig. 9-14, Table 2, 3 of the revised Supporting Information.

Responses to the report of the Reviewer 3

Comment 1. *Simultaneous evolution of H₂ and O₂ from photocatalysis remained a challenge in the field of photo-driven water-splitting. Theoretical calculation can only act as a supporting role in this research area, and a preliminary experimental study, at least, is essential to justify interpretations derived from theoretical calculations. If the authors' calculations are practically applicable, all the photocatalyst nanoparticles, with suitable band positions, dispersed between two GO layers can provide the same functions as those given by C₃N₄. The reviewer strongly suggests the authors to ask supports from experimentalists for justification of their calculations.*

Response: The reviewer has raised a philosophic question that is difficult to address. We could only present our own viewpoint on this issue. In the history of science, we have witnessed so many breakthroughs from “simple” theoretical predictions. The correctness of a theoretical prediction is justified by the rigorosity of the theoretical model and calculations. Its usefulness should of course be verified by experiments as the next step. At this stage, we want to tell the world a new way of designing photocatalytic systems for water splitting that comes out from the rational thinking and rigorous calculations. We are confident that the experimentalists will be able to utilize our idea in the near future, but this fact should not prevent the acceptance of our paper for

publication. We would like to use several examples to highlight the importance of theoretical predictions for the field of photo-driven water-splitting.

- (1) Despite of enormous experimental efforts, the simultaneous evolution of H₂ and O₂ remains a grand challenge. The complex interplay related to material design, fabrication and utilization is difficult to disclose experimentally, while first-principles calculations are able to bridge atomic-scale structures and properties, as well as the macroscopic functionality. With rapid progress in high performance computing, Tuan Anh Pham, Yuan Ping and Giulia Galli recently reviewed the key role of modelling heterogeneous interfaces for solar water splitting (Nat. Mater., 2017, 16, 401).
- (2) Exploring new architecture for solar-driven water splitting by the theoretical modeling can cast new ideas on the material design towards much improved performance. For instance, in a theoretical work of Du et al (J. Am. Chem. Soc., 2012, 134, 4393, see Ref. 14) a 2D hetero-structure of C₃N₄-Graphene for solar harvesting was proposed, which inspired the fabrication of the C₃N₄-carbon quantum dots by Kang *et. al.* (Science, 2015, 347, 970) and graphitic carbon nitride nanosheets by Ajayan *et. al.* (Adv. Mater., 2013, 25, 2452) that all exhibited excellent photocatalytic performance for hydrogen evolution.
- (3) As a theoretical group, it is impractical for us to perform in-depth experimental studies. It would take tremendous time to find a viable experimental collaborator to scrutinize our idea in realistic devices. In contrast, we believe that a publication in this influential magazine can definitely stimulate many experimental scientists to utilize their advanced technology and conditions for materializing this design in practical applications.
- (4) Meanwhile, we fully agree with the reviewer that our design is not limited to the graphene oxide and C_xN_y. It would be great to extend the concept to other semiconductor photocatalysts capsuled by graphene, fullerene, carbon-nanotube. Furthermore, we have added other functional groups on the outer graphene surface to promote water splitting efficiency (Supplementary Fig. 10-14 and Table 2, 3). We have thus added those outlook and new results of functional groups into the revised manuscript. Please refer to the red texts in Line 9-11/Paragraph 2/Page 10, and Line 3-15/Paragraph 2/Page 7 and Line 1-4/Paragraph 1/Page 8.

In fact, we are as confident as the reviewer to believe that the experimentalists will quickly utilize our new design. The quicker our paper is published, the earlier the real products will be produced.

REVIEWERS' COMMENTS:

Reviewer #2 (Remarks to the Author):

The authors have significantly revised the manuscript addressing the most important points of criticism. I recommend publication.